# Effects of Injector Nozzle Number of Holes and Fuel Injection Pressures on the Diesel Engine Characteristics Operated with Waste Cooking Oil Biodiesel Blends

**Mukur Beyan Ahmed and Menelik Walle Mekonen ***

College of Engineering, Ethiopian Defence University, Bishoftu P.O. Box 1041, Ethiopia; mukur2014@gmail.com
* Correspondence: menelikiitg@gmail.com

**Abstract:** This work covers the impact of varying injector nozzle hole numbers (INHNs) and fuel injection pressures (IPs) on fuel atomization, performance, and exhaust emission characteristics of a diesel engine. The primary goal of this research was to improve fuel characteristics. Increasing INHNs and fuel IPs have a substantial impact on the blended fuel viscosity and density, which leads to increased atomization and mixing rates, as well as combustion and engine efficiency. The fuel atomization was checked by varying the INHNs with an operating diesel fuel using the ANSYS Fluent spray simulation work. The experimental test was performed on the fuel blends of waste cooking oil (WCO)–diesel blends from 10 to 30% (with an increment of 10%) by evaluating the performance and emission parameters. The fuel IPs were altered on four, such as 190, 200 (default), 210, and 220 bar with a modification of INHN of 1 (default), 3, and 4), each 0.84, 0.33, and 0.25 mm in orifice size, respectively. The simulation result shows that the INHN-4 has better fuel atomization. Whereas the experimental test revealed that the increment in blending ratio of WCO was up to 30%, INHNs and fuel IPs enhanced the BSFC and BTE and reduced exhaust emissions. The results indicate that increasing the fuel IP up to 210 bar with a 4-hole INHN for B30 was the optimal combination for the overall enhancement of BSFC and BTE, as well as lower CO and HC emissions with a minor rise in $NO_x$ when compared to the baseline diesel.

**Keywords:** diesel engine; fuel injection pressure; injector nozzle number of holes; waste cooking oil

## 1. Introduction

Due to their better thermal efficiency, operational dependability, and resilience, diesel engines are widely employed and dominate power sources for the road transport industry [1]. Diesel engines, on the other hand, produce a lot of soot due to the non-homogeneous mixing of fuel and air. The performance and emission characteristics of CI engines depend on the injector nozzle fuel atomization and spray performance. The necessary parameters for the process of fuel atomization and combustion in an engine that controls the air–fuel mixing are the inner nozzle flow and spray character. The inner nozzle flow and spray structure are expected to be significantly altered and, consequently, the performance and emission features of the diesel engine due to differences in the physical properties of biodiesel and diesel fuel [2]. The fuel injector nozzle is one of the important components of the diesel engine. The nozzle hole numbers, increasing the injection pressure, and orifice sizes severely influence the combustion and performance due to the spray parameters such as penetration length and droplet size [3]. The fuel particle sizes will grow smaller as injection pressure is raised. The engine performance will improve when the formation of the fuel–air mixture improves during the ignition period. The ignition delay period shortens if the injection pressure is too high. As the likelihood of homogenous mixing declines, so does the efficiency of combustion. As a result, smoke is produced from an engine's exhaust. The injector's nozzle geometry is critical for managing diesel spray atomization and combustion. The nozzle hole size must be lowered to produce smaller droplets in order to minimize the

size of fuel droplets [4,5]. Biodiesel is usually used as a renewable, clean source due to its promising ability to reduce soot emissions in diesel engines. Accordingly, it is vital to examine the effect of the fuel injector nozzle on the performance and emission parameters with different types of fuel [3,6]. By evaluating the performance parameters, combustion characteristics, and emissions of a diesel engine operated with preheated palm oil methyl ester, this study investigates the effects of varying fuel injection pressure from 188 to 224 bar (with a 12 bar increment), fuel injection timing (19° bTDC, 23° bTDC, and 27° bTDC), and injector nozzle hole number (3, 4, and 5 holes), each 0.3 mm in diameter. The optimum combination for overall enhancement in BSFC and BTE with reduced emissions of CO and HC was a higher fuel IP of 212 bar and an advanced IT of 27° bTDC with a 4-hole INHN for warmed POME (114 °C) [4]. At 200, 220, and 240 bar injection pressures, the DI diesel engine's performance and emissions were evaluated using different nozzle hole sizes, such as 3 holes of 0.28 mm and 5 holes of 0.20 mm. The 5-hole 0.2 mm nozzle displayed improved performance and emission characteristics at a fuel injection pressure of 220 bar [4]. With the engine running at a constant speed of 1500 rpm and a CR of 17.5, the performance of the diesel engine was evaluated using neem oil methyl ester as fuel. The 3, 4, and 5 holes of 0.3 mm orifice size are used in the injectors. For testing, fuel injection pressures of 205, 220, 230, 240, and 260 bars were used, as well as fuel injection timings of 19°, 23°, 27°, and 31° bTDC. With neem oil methyl ester, it was discovered that 240 bar injection pressure, 27° injection time, and a 5-hole injector nozzle gave a higher performance [5]. Variable fuel injection timing of 19°, 23°, and 27° bTDC, as well as fuel injection pressures of 210, 220, 230, and 240 bars, and 3-, 4-, and 5-hole injectors of 0.3 mm in diameter were used to investigate the performance of a single-cylinder diesel engine. The results show that using a 4-hole injector with a 19° injection timing and a 230 bar injection pressure improves engine performance and reduces emissions [4,7]. Jatropha, which is not used for human food, was utilized to test the performance of a diesel engine. The tests were carried out on nozzle injectors with 5, 7, 9, and 11 holes and a 210 bar injection pressure. The 9-hole nozzle is thought to provide superior performance and lower emission rates [8]. The purpose of the study was to assess a three-hole nozzle by employing various diameters to see how varying nozzle hole sizes affect performance, combustion, and emissions. The diameters chosen were 0.28 mm for the base and 0.20 mm for the modified. In a short amount of time, the 0.20 mm modified nozzle was shown to accelerate vaporization, atomization, and air–fuel mixing [9].

The effect of alternative fuels on spray, performance, and emission characteristics of diesel engines, which influence the performance and emission parameters of diesel engines, is also subject to restricted research. On the other hand, many researchers have looked at the impact of varying the number of holes in injector nozzles on diesel engine performance and emission metrics. According to the available literature, no research has yet been conducted to determine the impact of INHNs and fuel IPs on a diesel engine running on waste cooking oil and biodiesel mixes. The current study considers various INHNs and fuel IPs while examining diesel engine characteristics. A computer model is used to investigate the impacts of different injector nozzles on fuel atomization, vaporization, and mixing rate formation. Furthermore, an experiment was carried out to see how they affected the performance and emission characteristics of diesel engines. The size of the nozzle and its holes, as well as the cylinder dimension, are taken into consideration while designing the nozzles (existing and modified) and the spray domain of the single-cylinder diesel engine. Based on the nozzle fluid flow domain and spray parameters on the spray domain in ANSYS Fluent, the computational model is used to assess the fluid flow inside the nozzles.

## 2. Methodology

Various numerical simulations were undertaken in the current work to achieve the objectives defined based on the identified research needs as described in this section.

### 2.1. Simulation Setup

The three main simulation stages in ANSYS Fluent are preprocessing, solver, and post-processing. In this study the only focus of the simulation was the injector head and combustion chamber design of the fuel injector nozzle. Figure 1 indicates the overall simulation and analysis flow chart. Figure 2 shows the schematic view of fuel injector nozzles that have single, 3, and 4 holes, diameters of 0.84, 0.33, and 0.25 mm each other. Figure 3 indicates the half sectional schematic view of the spray domain used for the simulation work. The half section from the overall geometry modeling used was in order to reduce the simulation time and satisfactory analysis. All simulation works in this research work were performed using DELL Laptop computer, core i-7, 8GB RAM, and 2.4 GHz processer (DELL, Inc., Austin, TX, USA).

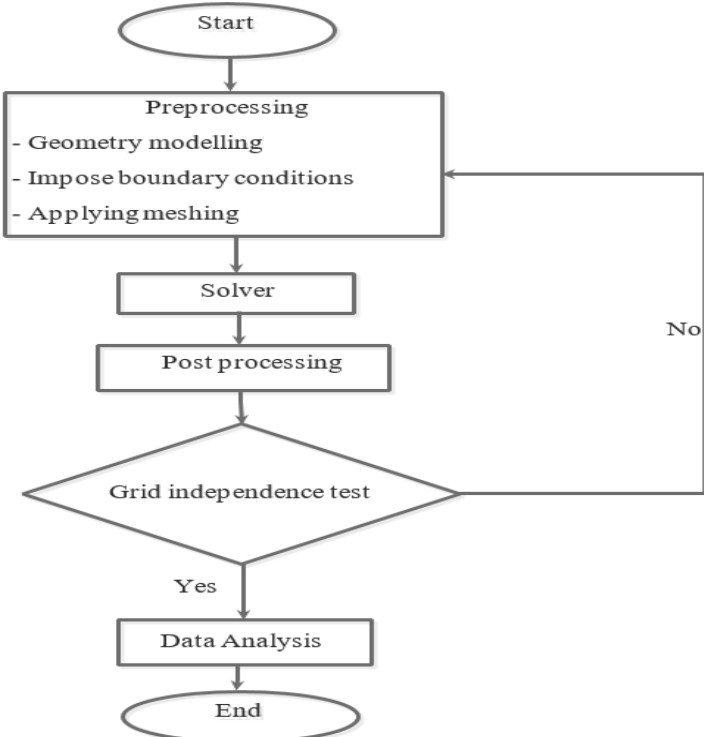

**Figure 1.** Modeling flow chart in CFD analysis.

### 2.2. Computational Models

The injector nozzles and their spray domain were modeled using the existing modeling software Solid works 2016. The commercially available package ANSYS Fluent 16.0 was set up with the three-dimensional (3D) calculation mesh of the nozzle and spray domain. The 3D injector nozzles and their spray domain are more reliable with the configuration of the actual profile, as computing resources authorization. Again, the 3D injector nozzles are convenient for tracking the velocity scattering in the various planes of the nozzle and the spray domain.

A 1-, 3-, and 4-hole diesel fuel injector with a nozzle angle of 150° was chosen based on their measurements (for 3 and 4 holes). To minimize computation time, only a single nozzle from the uniformly dispersed injector was modeled out of the three-hole and four-hole nozzles. The fluid flow domain inside the nozzle is depicted in Figure 2, and the primary difference between the models is their dimensions, which are based on the nozzles' technical specifications. The spray domain model was used in this investigation, and half of the model was chosen for all simulations to save computing time. The spray domain of the nozzles was modeled at the upper center of the domain in Figure 3, and the difference between the models was the nozzle specification.

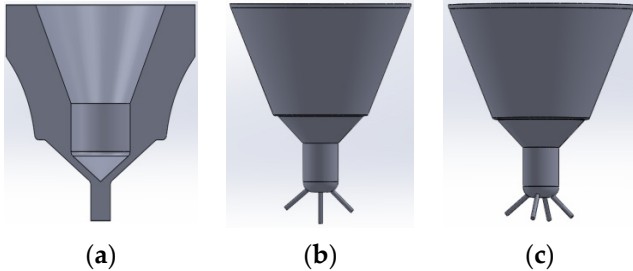

**Figure 2.** A diagram of fluid in nozzles: (**a**) single hole, (**b**) 3-holes, (**c**) 4-holes.

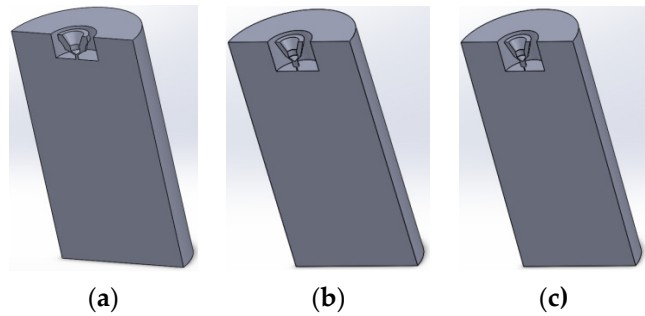

**Figure 3.** Computational fuel spray domains shown schematically: (**a**) single hole, (**b**) 3-holes, (**c**) 4-holes.

*2.3. Grid Independence Tests*

In order to accomplish a CFD analysis initially mesh must be applied to the geometry modeling that meets certain criteria in order to be successfully recognized by the analysis program. The basic parameter for ANSYS Fluent mesh is the maximum cell skewness that must be under 0.98, which is the basic concern after the mesh has been generated [10]. The simulation analysis was valued through the range of meshes from low to extra fine. The optimum mesh from the data analysis was evaluated by the mesh resolution that contained an error within a 5% limit [11–13]. Figure 4 indicates the grid independence tests of the overall simulation analysis models.

For the tested nozzles, five sets of meshes with different refinement levels were established in this study; the number of elements in each set was 133416, 205940, 345269, 392647, and 408662 (for INHN-1), 125345, 141006, 158186, 202844, and 245541 (for INHN-3), and 116985, 133908, 140095, 170946, and 211568 (for INHN-4) The velocity of the nozzle exit was used as one of the major characteristics in the numerical simulation of the inner flow of the nozzle, and the velocity parameters were estimated using five different meshes. The flow velocity increased consistently with mesh refinement, as shown in Figure 4a. The flow velocity increases dramatically as the number of pieces increases. The flow velocity tends to be stable when the number of nodes reaches its maximum. The Grid Convergence Index (GCI) of the nozzle with 408662 (for INHN-1), 245541 (for INHN-3), and 202844 (for INHN-4) elements is 2.97%, 1.72%, and 1.24%, respectively, using the GCI method to estimate mesh discrete error. Similarly, spray tip penetration was used as a characteristic parameter in the fuel spray simulation. When the element number of the spray domain is 560172, 704079, and 730792 for INHN-1, INHN-3, and INHN-4, respectively, the computed GCI was 2.64%, 2.29%, and 1.58%, which may match the mesh convergence criterion as shown in Figure 4b–d.

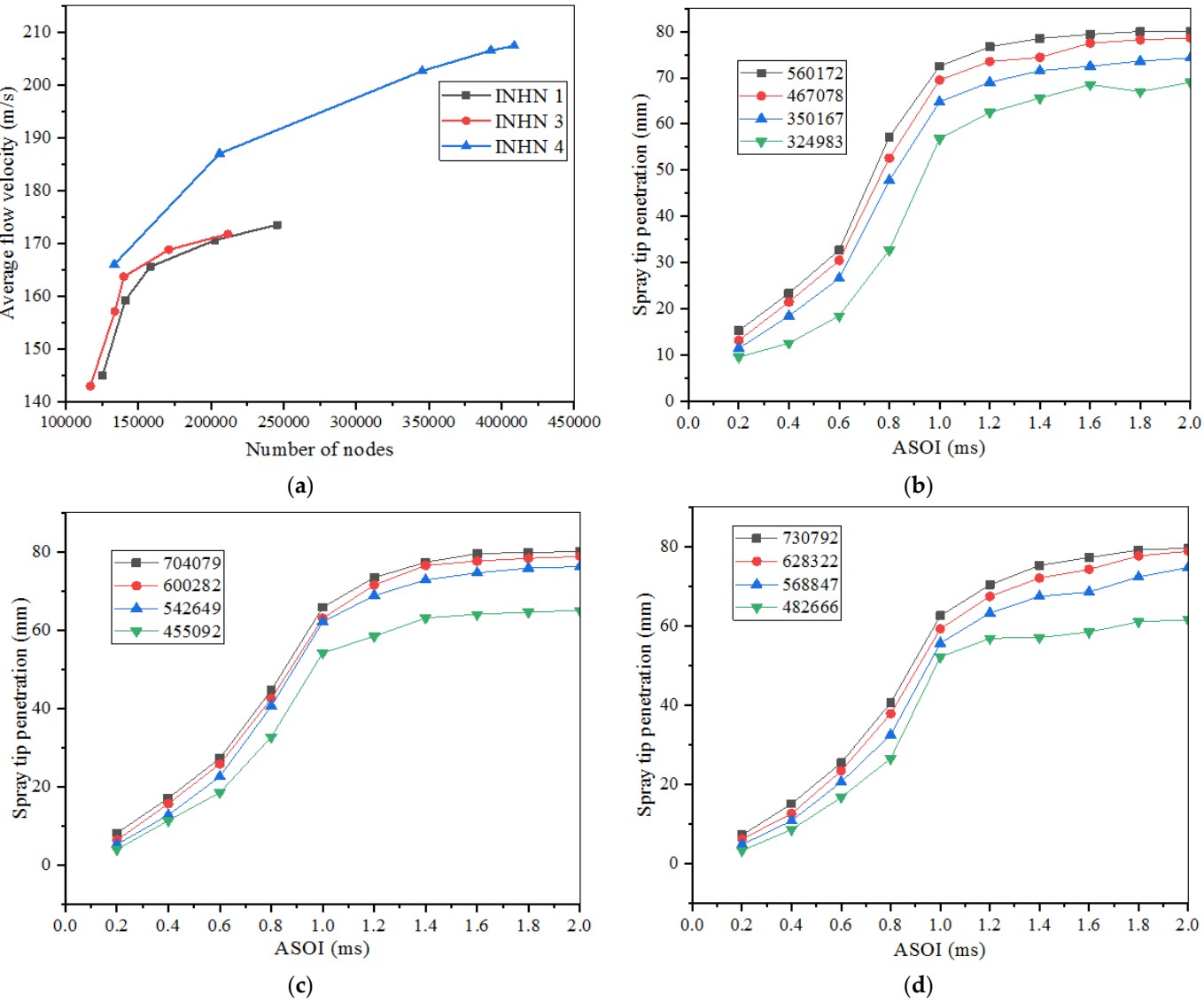

**Figure 4.** Spray simulation grid independence test: (**a**) nozzle models, (**b**) INHN-1, (**c**) INHN-3, and (**d**) INHN-4.

### 2.4. Boundary Conditions

In the numerical modeling of nozzle internal flow, the inlet and outflow of the nozzle were adjusted to pressure boundaries. As illustrated in Table 1 and Figure 5, the nozzle's inlet pressure was set to fuel injection pressure, the outlet pressure was set to ambient pressure, and the nozzle's wall surface was set to a no-slip velocity boundary.

**Table 1.** Boundary conditions for operation.

| Boundary Conditions | Value and Unit |
|---|---|
| Injection pressure | 200 bar |
| Back pressure | 2 bar |
| Ambient air temperature | T = 300 K |
| After start of injection (ASOI) | 0–2 ms |

In the numerical simulation of diesel spray characteristics, the top surface, bottom surface, and circumferential surface of the spray domain were all set to the fixed wall boundary. Furthermore, the bottom surface's boundary type was set to "escape", and the

surrounding wall's boundary type was set to "reflect". First, the transient flow inside the nozzle was numerically calculated. The spray numerical computation was then started with the calculation results at the nozzle outlet, which improved the accuracy of the spray simulations.

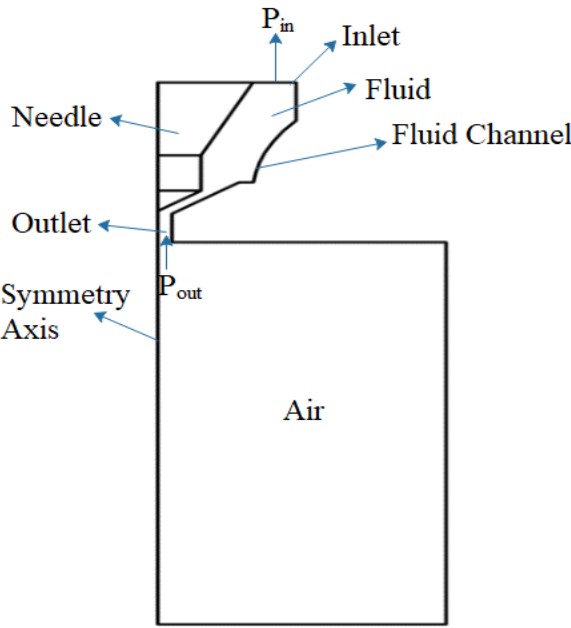

**Figure 5.** Schematic view of spray simulation boundary conditions.

*2.5. Mathematical Model and Governing Equations*

The mathematical model and governing equations utilized in the nozzle flow and spray domain simulations are discussed as follows:

(a)   Injector nozzle flow phase

The numerical calculations in this study were performed using ANSYS Fluent version 16.0 CFD software. The Eulerian multi-phase flow is used for the 3-D simulation of spray characteristics. The multiphase flow was described using the mixture model, which considered that air and liquid phases were mixed uniformly. Eulerian model is used to model the continuous flow of particles based on the Navier–Stokes equation. The Navier–Stokes governing equations solve the continuous flow of the fuel in the injector nozzle consisting of the continuity, momentum, and energy equation, which are listed in Equation (1) below [14].

$$\frac{\partial \rho}{\partial t} + \frac{\partial}{\partial x_i}(\rho u_i) = 0$$

$$\frac{\partial \rho}{\partial t}(\rho u_i) + \frac{\partial}{\partial x_i}(\rho u_i) = -\frac{\partial P}{\partial x_i} + \frac{\partial \pi_{ij}}{\partial x_j} + \rho g_i + F_i \tag{1}$$

$$\frac{\partial \rho}{\partial t}(\rho h) + \frac{\partial}{\partial x_i}(\rho u_i h) = -\frac{\partial P}{\partial x_i} + \frac{\partial}{\partial x_j}\left(k\frac{\partial \pi_{ij}}{\partial x_j}\right) + \eta \gamma$$

The Shear Stress Transport (SST) k-solver was used for all of the turbulence model research detailed in this work because it provides advantages for boundary layer problems and is more exact and resilient than other turbulence models. As stated in Table 2, the diesel fuel property employed in this analysis was based on values measured and learned in previous investigations.

**Table 2.** Properties of diesel fuel ued in spray simulation [13].

| Property | Value and Unit |
|---|---|
| Density | 835 kg/m$^3$ |
| Kinematic viscosity | 3.9 mm$^2$/s |
| Dynamic viscosity | 3.2 mPas |
| Surface tension | 0.0281 N/m |
| Heating value | 4.31 MJ/kg |

(b) Discrete Particle Phase Model

The discrete particle model (DPM) was utilized to capture the flow of vapor formation and atomization throughout the spray development simulation. To compute the continuous phase, the Navier–Stokes equations and the continuity equation were solved, while the force balance on the particle was integrated into the solver to determine particle trajectories. The particle trajectories were estimated using the result of the continuous flow for two-way interaction, whereas the continuous phase was determined using the Eulerian model [10,15]. Figure 6 depicts the two-way connection of the process scheme.

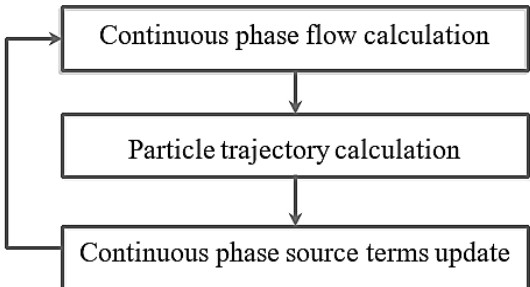

**Figure 6.** Coupling scheme [11].

Using a VOF-DPM (volume of fluid—discrete particle model) hybrid technique, as well as KH–RT (Kelvin Helmholtz–Reynolds Transport) and the TAB (Taylor Breakup) model, this study explores the effect of INHNs and diameter size on particle penetration, particle velocity, and particle mean diameter. A VOF-DPM hybrid approach is an Eulerian-Lagrangian approach that solves the Naiver-Stokes and other governing equations by treating the continuous liquid phase as a continuum. The precision of the droplets (dispersed phase) is determined by how effectively the interface is recorded by the VOF model, as well as the accuracy of the droplet identification technique. The ANSYS Fluent droplet identification algorithm facilitates the transition of a lump from the Eulerian to the Langrangian (DPM) phase. For spray simulations, a similar strategy was utilized [16].

### 3. The Experimental Setup and Procedures

A single-cylinder, 4-stroke DI-CI engine rated at 5.67 kW powers the experimental test setup. A hydraulic dynamometer serves as a loading system, as well as a fuel supply system, water-cooling, lubrication, and exhaust emissions. Figure 7 depicts a schematic illustration of the experimental setup. The setup allows for the evaluation of diesel engine performance metrics and exhaust elements. BP, BSFC, and BTE are some of the engine performance measures. A test engine's technical specifications are shown in Table 3. An FGA-4100 automobile emission analyzer is used to measure the levels of engine exhaust emissions. During each run of the engine, the CO and $CO_2$ emissions were monitored in percentage volume (% vol), whereas the HC and $NO_x$ emissions were measured in parts per million (ppm).

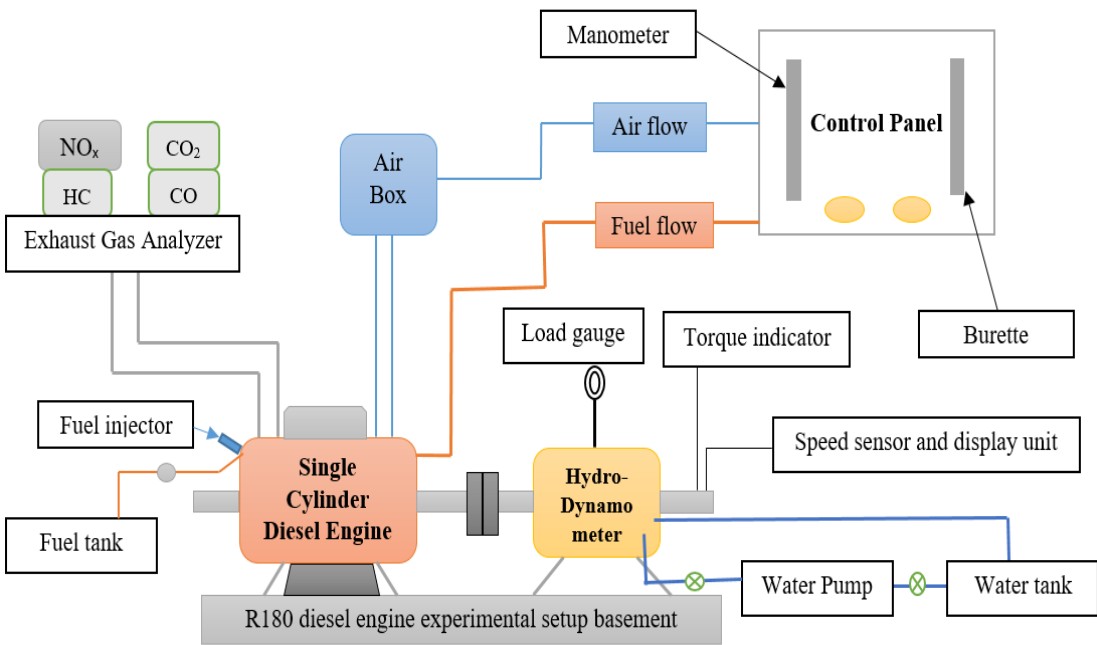

**Figure 7.** A diagram of the experimental setup.

**Table 3.** The test engine's technical specifications.

| Engine Type | R180, Single-Cylinder, Four Stroke Diesel Engine |
|---|---|
| Make of mode | R180 |
| Loading device | Hydro-dynamometer |
| Rated power | 5.67 kW at 2600 rpm |
| Cylinder bore and stroke | 80 mm × 80 mm |
| Displacement volume | 402 cc |
| Compression ratio | 21:1 |
| Injection timing | 23° bTDC |
| Nozzle type, hole diameter | Single hole, 0.84 mm |
| Nozzle opening pressure | 200 Bar |

### 3.1. Fuel Properties

The test fuel quality attributes that fuel should have when utilized in a diesel engine are shown in Table 4. The quality of the fuel, as well as the performance and emission parameters of a diesel engine, are all affected by fuel attributes [17]. The fuel characteristic values must be in the range of the most well-known biodiesel standards, such as ASTM D6751 and EN14214, in order to set limitations and fulfill standard diesel engine parameters. Table 4 shows a basic analysis of the existing B10, B20, B30, and diesel fuel characteristics. For testing, B10 (10% biodiesel + 90% diesel), B20 (20% biodiesel + 80% diesel), B30 (30% biodiesel + 70% diesel), and D100 (100% diesel) would be the fuels produced. The measured calorific values are comparable to those of diesel fuel. To avoid problems with the fuel atomization and spray characteristics, each biodiesel must meet the minimum standard limitations (EN 14214 and ASTM D6751) for diesel engine application [5].

**Table 4.** The properties of the test fuels that were employed.

| Properties | Diesel | WCO | B10 | B20 | B30 |
|---|---|---|---|---|---|
| Density at 27 °C (kg/m$^3$) | 835 | 913.6 | 847.5 | 853.6 | 860.4 |
| Calorific value (MJ/kg) | 43.1 | 37.28 | 41.56 | 40.82 | 39.5 |
| Kinematic viscosity at 40 °C (mm$^2$/s) | 3.16 | 10.63 | 6.72 | 5.56 | 4.25 |
| Flash point (°C) | 96 | 223 | 174 | 156 | 113 |
| Cetane number | 48 | 53.3 | 51.8 | 49.6 | 48.8 |

### 3.2. Test Procedures

The experimental analysis was carried out on the following three distinct types of injector nozzle hole numbers: 1-hole 0.84 mm, 3-hole 0.33 mm, and 4-hole 0.25 mm, all of which were fueled with WCO biodiesel fuel blends. The technical specifications of injector nozzles used in this study is tabulated in Table 5. The experiments were carried out using the following engine operating parameters of a fixed compression ratio of 21:1 with 23° bTDC fuel injection timing under varying nozzle opening pressure of 190, 200, 210 and 220 bar, respectively. During executing the experiment, the fuel injection pressure was varied by altering the injector spring tension (Figure 8). The photographic view of the nozzles used is shown in Figure 9. The engine was run at various loads to produce the best operating load state for performance and emissions testing, including no-load, 20%, 40%, 60%, 80%, and 100% load. Sufficient attention was given to load the engine precisely at each phase of load and to sustain ambient conditions constant, in which each test was cross-checked on different days. In each of the sets of experiment, the readings of the fuel flow, CO, $CO_2$, $NO_x$, and HC emissions were noted. In which the fuel flow rate and air flow rate were recorded using a fuel burette and a u-tube manometer, respectively, and the various emissions concentrations, such as CO, HC, $CO_2$, and $NO_x$ emissions, were also measured using an exhaust gas analyzer. While the performance parameters of BP, BSFC, and BTE were calculated using Equation (2) based on [18]. In the end, the diesel engine performance and emission parameters were plotted against varying engine load and then varying INHNs and fuel IPs, respectively. In the test matrix, Table 6 displays the details of the experiment.

$$\begin{aligned}
\text{BP} &= \tfrac{2\pi \times \text{N} \times \text{T}}{60,000} \ [\text{kW}] \\
\text{BSFC} &= \tfrac{\dot{m}_f \times 3600}{\text{BP}} \ [\text{kg/kW·h}] \\
\text{BTE} &= \tfrac{\text{BP}}{\dot{m}_f \times \text{Q}_{\text{LHV}}} \ [\%]
\end{aligned} \tag{2}$$

**Table 5.** Details of the injector nozzles.

| Model | Nozzle Label | No. of Holes | Diameter of Nozzle Orifice Hole (mm) |
|---|---|---|---|
| ZS4S1 | INHN-1 (existing) | 1 | 0.84 |
| DL 90S3 | INHN-3 (pd-1) | 3 | 0.33 |
| DL 100S1053 | INHN-4 (pd-2) | 4 | 0.25 |

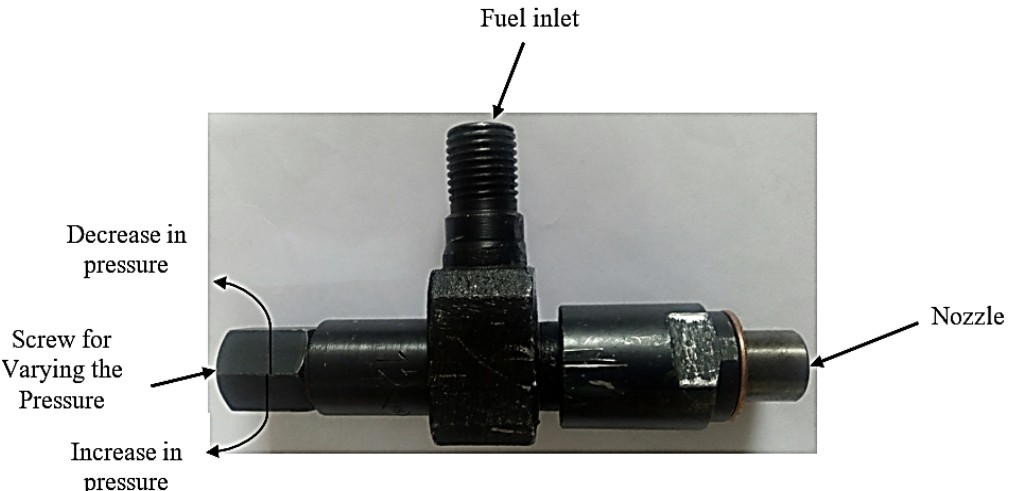

**Figure 8.** R180 diesel engine fuel injector assemblies.

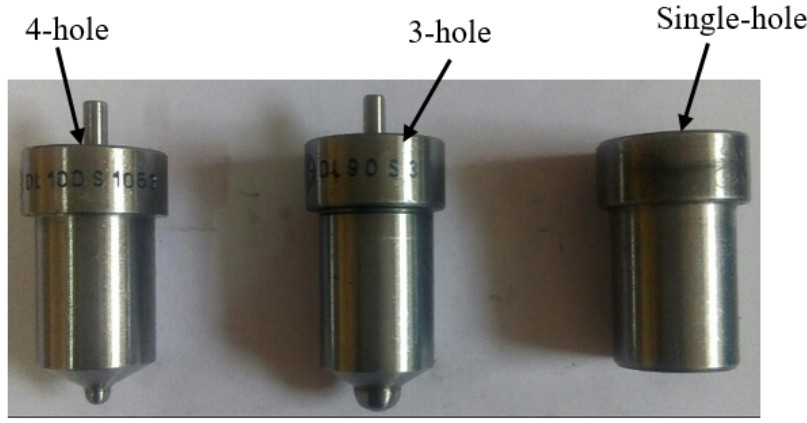

**Figure 9.** Different injector nozzles photograph.

**Table 6.** The test matrix for the experimental analysis.

| Operating and Design Conditions | | | | |
|---|---|---|---|---|
| **Experiment** | **IP (Bar)** | **INHN** | **NHD** | **Test Fuels** |
| 1. Investigate the effects of varying engine load (no load, 20%, 40%, 60%, 80%, and 100%) and fuel blending ratio (100D, B10, B20, and B30) on engine performance and emission parameters. | | | | |
| Baseline data test | Default 200 | 1 | 0.84 | 100D, B10, B20, B30 |
| 2. Investigation of the influence of varying INHNs and fuel IPs a diesel engine running at an optimum engine load (80%) and better blending ratio (B30) of engine performance and emission parameters. | | | | |
| Experimental test | 190–220 with 10 bar increments | 1, 3, and 4 holes | 0.84, 0.33, and 0.25 mm | B30 |

*3.3. Experimental Test Matrix*

The number of tests that were executed and guidance for experimental work is an experimental test matrix. The baseline and detailed experimental test matrices are listed below. From the experimental investigation observations, the superior combined combinations of the injector nozzles operating at optimal load conditions, INHNs and fuel IPs with better brake thermal efficiency, fuel economy, and reduced exhaust emission are obtained.

**4. Results and Discussion**

In this section, the results of inside nozzle flows and spray development in the chamber are presented. The test results on the performance and emissions of different INHNs and fuel IPs executed with WCO–diesel blend fuel are also discussed here.

*4.1. Fuel Spray Simulation Results*

(a)    Influence of INHNs and orifice size on the spray width

One of the parameters, which is the spray width, was taken into account to compare the spray characteristics of different nozzles. The data of the spray width were compared using the ratio of spray width over the nozzle hydraulic diameters. Figure 10 shows that the spray width of the INHN-3 and INHN-4 was significantly bigger than that of the INHN-1. These results revealed that the superior spray character with a greater spray width would be attained through the use of the INHN-4 nozzle compared to the INHN-1 and INHN-3.

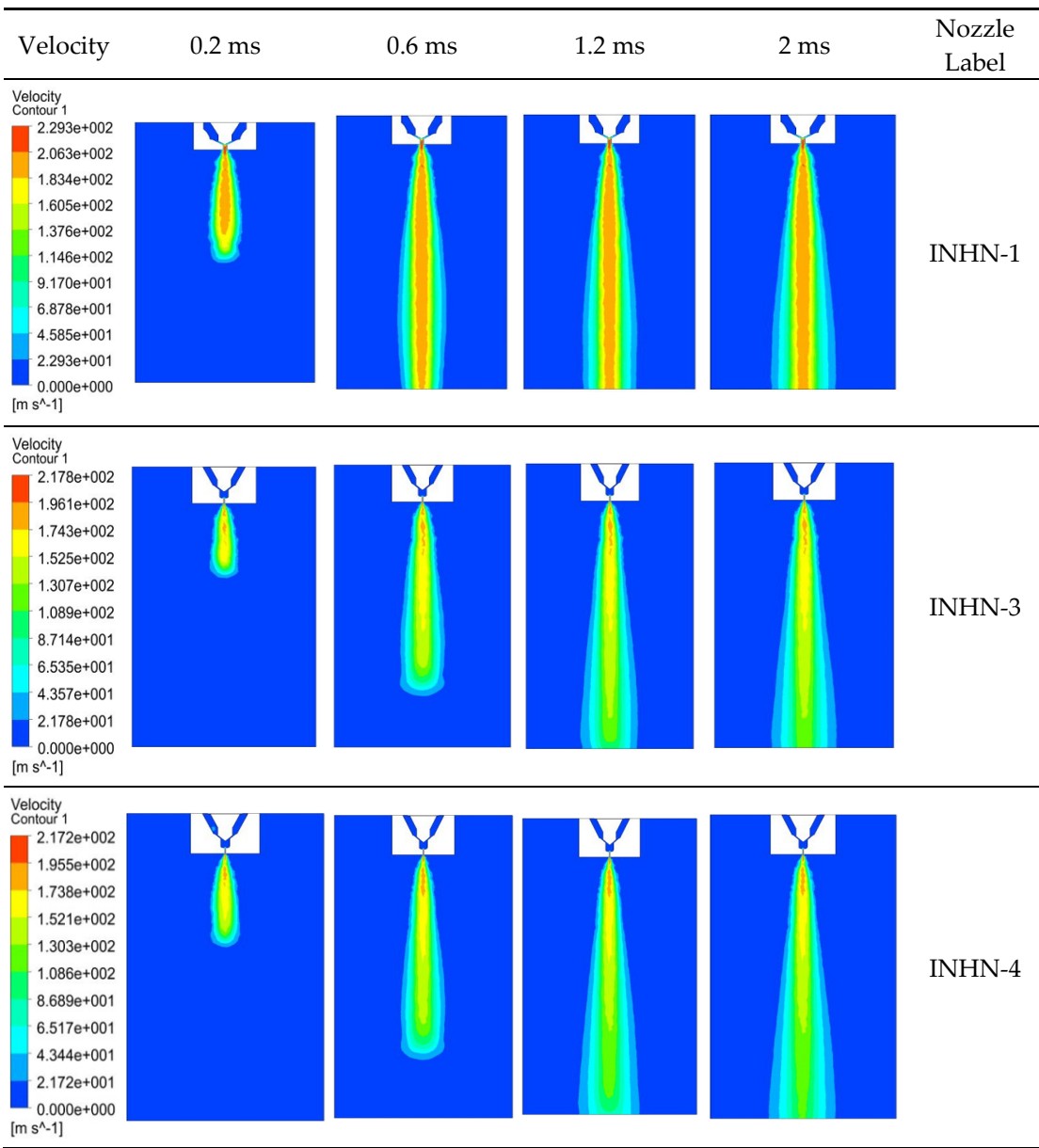

**Figure 10.** Spray domain model contour velocity for INHN-1, INHN-3 and INHN-4.

(b)    Influence of INHNs and orifice size on the particle penetration

In comparison to the INHN-3 and INHN-4, the INHN-1 had more aerodynamic drag, which resulted in a longer penetration. The spray penetration length increases progressively as the injector nozzle diameter grows, as expected because the increased pressure differential results in higher velocities and momentum at the nozzle exit, which aids drop penetration, as illustrated in Figure 11.

(c)    Influence of INHNs and orifice size on the spray particles velocity

Because of the higher pressure differential, velocity increases slightly when it enters the chamber at a 200 bar injection pressure. However, this rise was not seen in INHN-1, and INHN-3 velocities are higher than INHN-4 velocities at the same axial positions. The velocity declines rapidly at first, then increases, and then gradually decreases till the end, as illustrated in Figure 12.

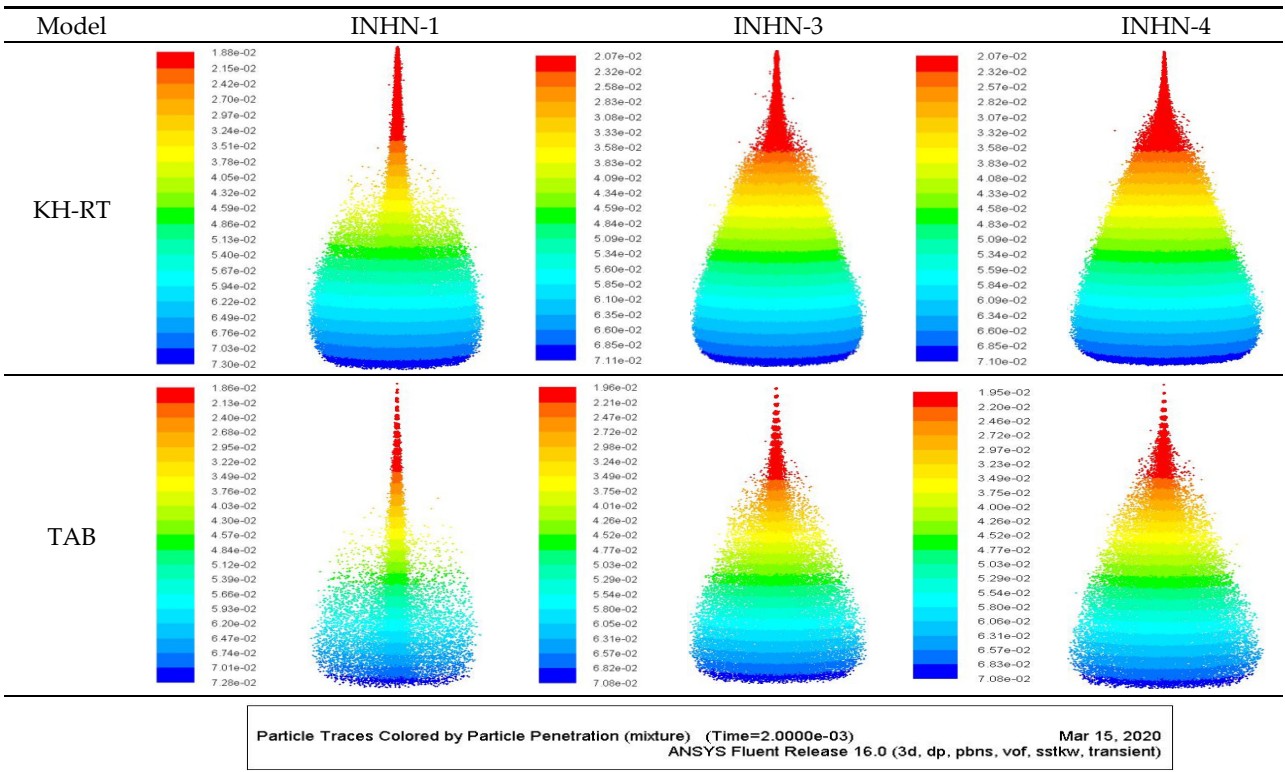

**Figure 11.** Particle penetration for different injector nozzles.

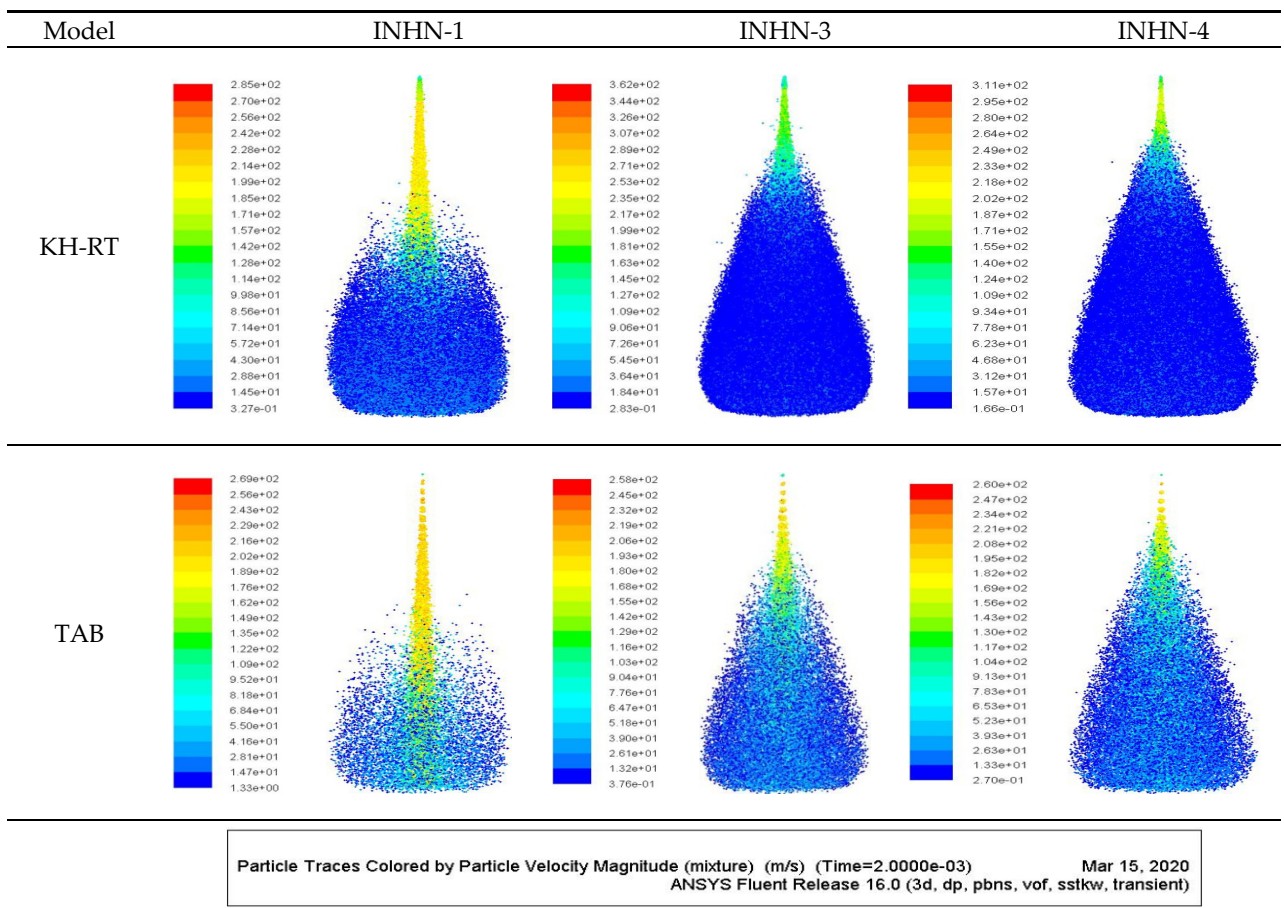

**Figure 12.** Particle velocity magnitude for different injector nozzles.

(d)    Influence of INNHs and orifice size on the particles mean diameter

The ratio of the drop volume to its surface area is the particle mean diameter [19,20]. By gradually reducing the droplet size difference while increasing INHNs and decreasing the particle mean diameter, consistent particle size distribution was achieved (Figure 13). Further experimental tests of the various injector nozzles based on varied injection pressures are required, as mentioned in all of the simulation work findings.

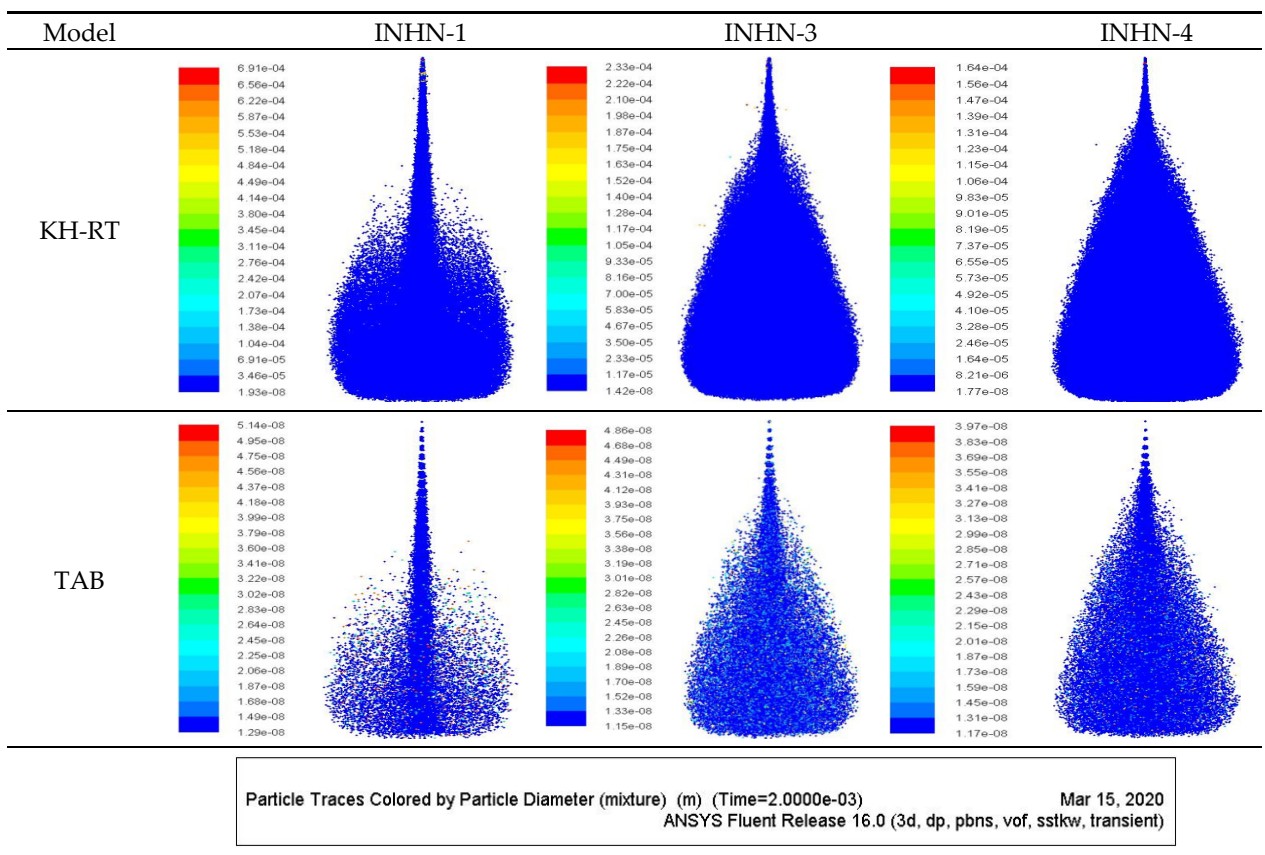

**Figure 13.** Particle diameter for different injector nozzles.

All of the simulation work results are an indication of the further experimental tests of the various injector nozzles based on different fuel injection pressures to gain a superior combined combination of fuel injector nozzle, fuel injection pressure, and a better blending ratio of WCO biodiesel operating blend fuel. These parameters of the diesel engine characteristics were not performed in the simulation analysis but were performed experimentally.

### 4.2. Experimental Results

The fuel characteristics and engine performance (BSFC and BTE) at optimum load circumstances were investigated using a waste cooking oil (WCO) mix fuel in the current study. The experimental findings were obtained by combining biodiesel fuels at up to three distinct blending ratios, ranging from 10% to 30% in 10% increments. Finally, the fuel properties were compared to the biodiesel specifications.

#### 4.2.1. Influences of Engine Load on Diesel Engine Performance and Emission Parameters

The engine performance parameters can be evaluated from the engine brake torque, brake power, brake thermal efficiency, and brake specific fuel consumption.

(a)    Effects on brake specific fuel consumption and brake thermal efficiency

Figure 14 illustrates that the brake thermal efficiency (BTE) and brake-specific fuel consumption have opposing trends (BSFC). It depicts the impact of engine load on BTE and BSFC. The trials were conducted in order to determine the optimal load size that would provide the best fuel economy and efficiency. The BTE increases with engine load but reduces significantly between 80 and 100% loads, as can be seen. This modest decrease could be attributed to a low extra air ratio at high engine loads, which harmed combustion [4]. In this situation, INHN-1, the BSFC, is reduced greatly, and the BTE is boosted to its maximum at an 80% load. The optimum load will be employed for the subsequent execution of the impacts of adjusting the fuel IPs and INHNs based on the outcome of the 80% load condition.

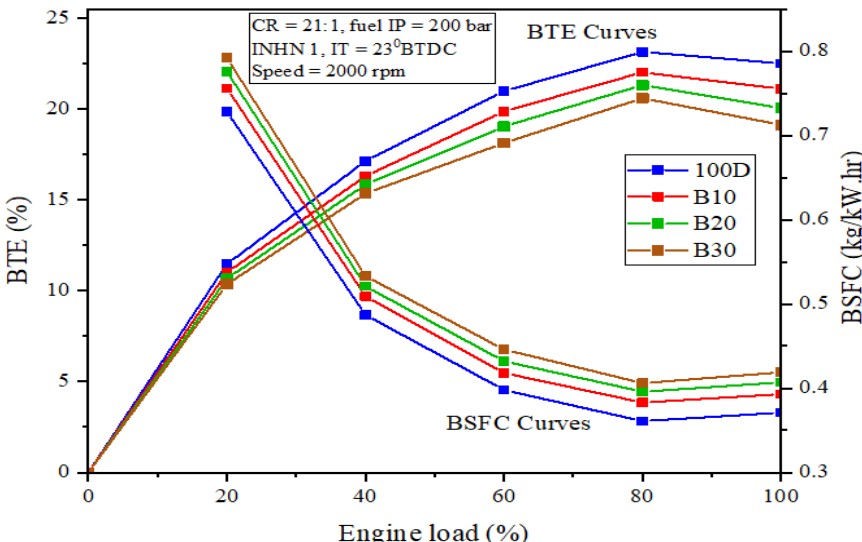

**Figure 14.** Influences of engine load on brake specific fuel consumption (BSFC) and brake thermal efficiency (BTE) for various test fuels.

(b)    Engine exhaust emissions analysis

The incomplete burning of fuels that do not include oxygen in their chemical structure produces carbon monoxide (CO). The CO emissions from a diesel engine running on diesel and WCO mixed fuels with various loads are shown in Figure 15a. The CO emissions were lowered at higher load conditions due to improved fuel atomization, vaporization, distribution, and mixing rate of injected fuel, resulting in better air–fuel mixing and the burning of more carbon fuel [21]. The CO emissions for INHN-1 at fuel IP 200 bar were shown to be lower when the load was increased from 0 to 100% [4]. The carbon dioxide ($CO_2$) emissions increased from 0% to 100% during load change as a result of starved atomization and incorrect combustion. The change in $CO_2$ emissions when diesel fuel is utilized with changing loads is depicted in Figure 15b. The trends for $CO_2$ emissions, on the other hand, are the polar opposite; $CO_2$ emissions increased as engine load increased. This could be due to the increased volume of intake air in the combustion chamber as the engine load increases, pushing it to use lean mixes that resulted in better combustion compared to low engine load situations [5].

Because of better atomization and combustion, the unburned hydrocarbon (HC) emissions for INHN-1 at fuel IP 200 bar reduced as the load increased from 0 to 100%. A shorter igniting delay will result from improved atomization. Higher temperatures in the combustion chamber, caused by correct atomization and evaporation, have reduced INHN-1 HC emissions by 0.84 mm. Figure 15c depicts the HC emissions produced by a diesel engine running on WCO–diesel blend fuel at various loads. This means that when the load increases, the HC emissions from the WCO–diesel blend fuel decrease. These

decreases imply that when the load is high, the fuel undergoes extra combustion, resulting in a significant reduction in HC emissions [4]. Because of the greater combustion phase, the nitrogen oxide ($NO_x$) emissions for INHN-1 at fuel IP 200 bar increased as the load was increased from 0% to 100%. The $NO_x$ emissions for the test diesel fuel with varied loads are shown in Figure 15d. The data reveal that as the load increases, so do the $NO_x$ emissions. As the load grows, the temperature of the combustion gas rises, increasing $NO_x$ emissions. This is because $NO_x$ production is affected by the temperature of the cylinder gas [7].

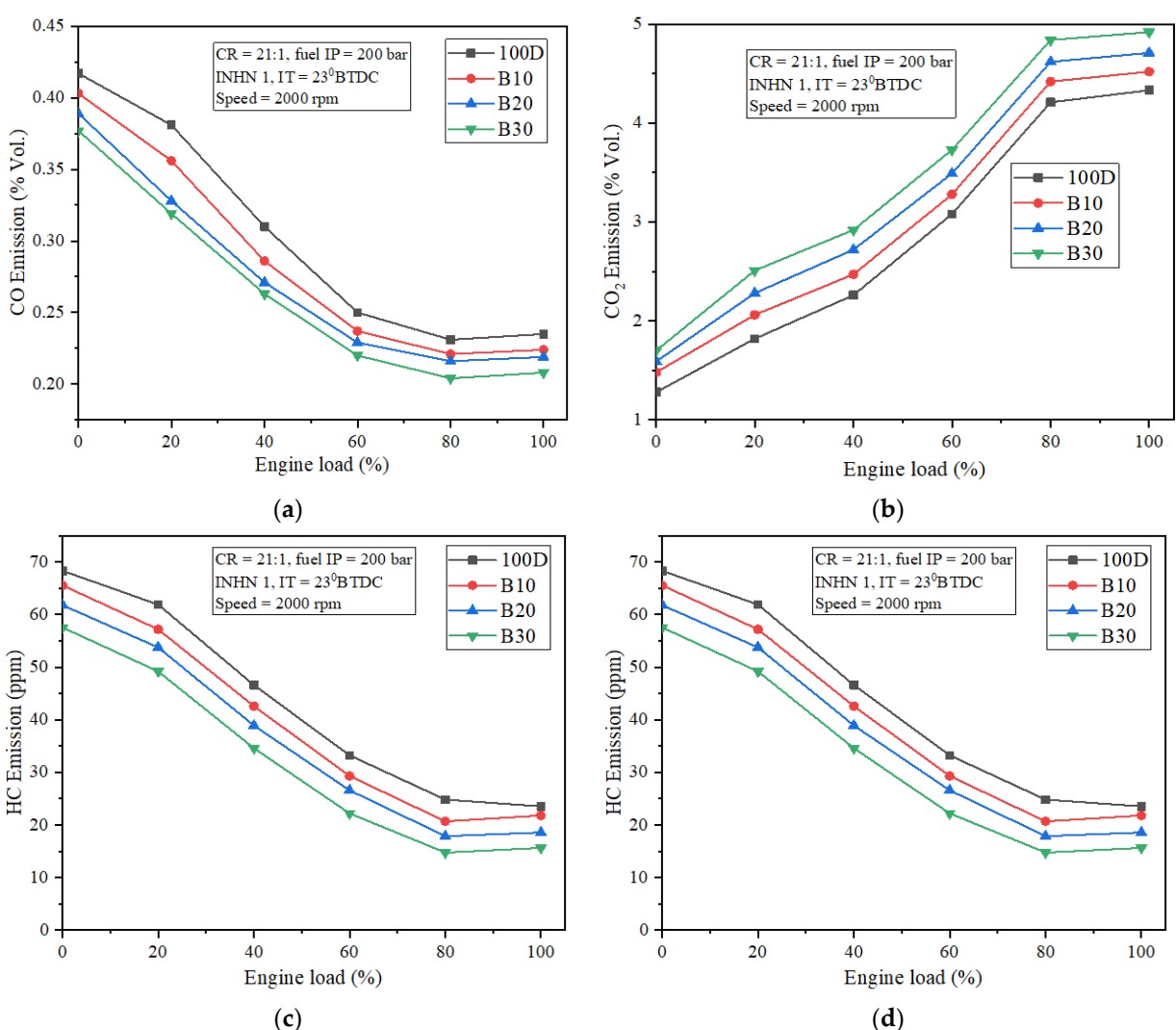

**Figure 15.** Influence of engine load on (**a**) CO; (**b**) $CO_2$; (**c**) HC; (**d**) $NO_x$ emissions from engine.

### 4.2.2. Influences of Injector Nozzle Number of Holes and Fuel Injection Pressures on Engine Performance and Emission Parameters

Fuel injection pressures (IPs) and injector nozzle number of holes (INHNs) are two critical parameters that affect a diesel engine's performance and emissions. In a diesel engine, the primary determinants of combustion are fuel spray formation and mixing rate. Increased fuel IP results in increased fuel atomization, better spray characteristics, and a shorter physical delay period, resulting in improved premixed combustion and quick combustion rates, as well as improved engine performance metrics and lower CO and HC emissions. A high fuel IP, on the other hand, produces delayed injection, which can result in a greater velocity of droplets going through without properly mixing the air, resulting in poor engine performance and higher emissions due to faulty combustion. The INHN and diameter size play a role in fuel spray atomization, evaporation, distribution, and

combustion. Increased INHNs result in better combustion by increasing air–fuel mixing, fuel vaporization, and heat release rates.

(a)    Effects of INHNs and fuel IPs on brake specific fuel consumption

Figure 16a depicts the influence of INHNs and fuel IPs on BSFC in an ideal load situation. The goal of the tests was to find the best injector nozzle number of holes and fuel injection pressure for optimal fuel economy. Because of the starved atomization, the BSFC increases as the nozzle hole diameter (NHD) increases. Figure 16a shows that for a fuel IP of 200 bar, the values of BSFC for INHN-1 are lower than for 190 bar, 210 bar, and 220 bar. Higher BSFC values have come by increasing the fuel IP beyond 200 bar. This could be because increasing the injection pressure reduces the size of the fuel droplets while also increasing their momentum. As a result, excessively large increases in pressure would have produced even smaller droplets, but the droplets' motion would have caused them to impact the cylinder's inner wall, resulting in increased fuel consumption to provide the same power. Because of greater rates of air–fuel mixing in INHN-3 and INHN-4, the BSFC decreases dramatically with increasing fuel IP until 210 bar. This indicates that a smaller diameter nozzle necessitates a higher injection pressure in order to achieve complete combustion and reduce fuel usage. As a result, in the current conditions, a fuel IP of 210 bar resulted in a reduced BSFC. It can be shown that the BSFC with INHN-3 and INHN-4 and a fuel IP of 210 bar has the lowest values; thus, it can be concluded from these studies that INHN-4 and a fuel IP of 210 bar yield lower values. Figure 16a depicts the impacts of fuel INHNs on the BSFC of WCO biodiesel (B30) blend fuel for various IPs at 80% operational load. At 210 bar fuel IP, the results reveal that BSFC dropped as INHN increased from one to three and four INHNs. This is due to the increased number of holes, which improved fuel atomization and increased the fine droplets of injected diesel. At INHN-4, the average drop in BSFC of diesel test fuel was 5.62% compared to INHN-3, for fuel IP of 210 bar. However, as compared to the initial fuel IP of 200 bar, the 210 bar fuel IP for all INHNs resulted in a drop.

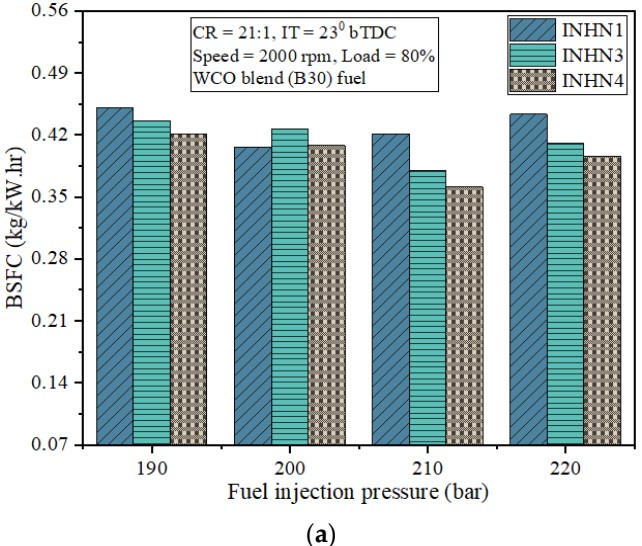 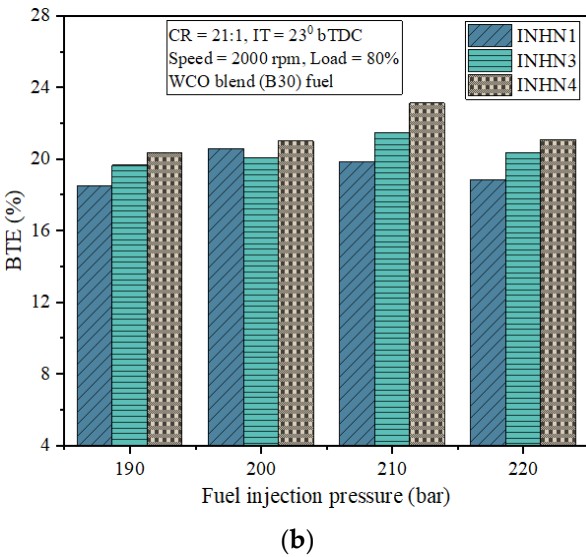

**Figure 16.** Influences of INHNs and fuel IPs on engine performance: (**a**) BSFC and (**b**) BTE.

(b)    Effects of INHNs and fuel IPs on brake thermal efficiency

Figure 16b depicts the influence of INHNs and fuel IPs on the BTE of a diesel engine at an optimal load state for WCO mix (B30) fuel. According to the findings of INHN-1, there is an initial increase in BTE with an increase in fuel IP from 190 to 200 bar, but a decrease in BTE with an increase in fuel IP from 210 to 220 bar. As a result, in the current conditions, a fuel IP of 200 bar resulted in a greater BTE for INHN-1, which is the best option. Figure 16b

shows that raising the fuel IP from 200 to 210 bar BTE increased the fuel IP by 10.36% and 10.96% at INHN-3 and INHN-4, respectively. This was due to an increase in INHN, which resulted in improved air–fuel mixing, fuel vaporization, and combustion and heat release rates. As a result, BTE rises as the number of INHNs rises. The smaller nozzle hole diameter also increases the mixing, which is evidenced by a shorter combustion duration. As a result, the heat and time losses are reduced, resulting in a greater BTE and a lower BSFC. One of the nozzles with a smaller diameter (Ø 0.25 mm) has enhanced combustion for all of the test situations, resulting in an increase in fuel conversion efficiency compared to the reference nozzle (Ø 0.84 mm).

(c)    Effects of INHNs and Fuel IP on Carbon Monoxide Oxide Emission

Figure 17a depicts the influence of INHNs and fuel IPs on CO emissions for WCO biodiesel (B30) blend fuel at an optimal load situation. CO emissions, as we all know, are the result of incomplete combustion caused by a rich air–fuel mixture. Furthermore, it was shown that as INHN and fuel IP increased, CO emissions reduced. This has the added benefit of increasing the fuel IP and injector nozzle of the hole to a certain extent. CO emissions are lower at fuel IP 210 bar and INHN-4 injector nozzle due to enhanced atomization and proper combustion. The CO emission for B30 with INHN-4 was found to be greater at a standard 200 bar fuel IP. In contrast, when the fuel IP was reduced to 190 bar, CO emissions rose with the identical INHN-4. However, when the fuel IP was increased to 220 bar, the CO increased again due to the mixture's non-uniform composition throughout due to a lack of the air entrainment essential for achieving a stoichiometric mixture [4]. When the fuel IP was increased from 200 to 210 bar using INHN-4, the CO emission level dropped by 4.99%. However, when compared to 200 bar, CO emissions for INHN-4 increased by 3.48% and decreased by 3.52% at 190 pressure and 220 bar fuel IPs, respectively.

(d)    Effect of INHNs and Fuel IP on Carbon Di-Oxides Emission

Figure 17b shows the influence of INHNs and fuel IPs on $CO_2$ emissions at an ideal load situation for WCO biodiesel (B30) blend fuel. $CO_2$ emissions, as we all know, are the result of incomplete combustion caused by a rich air–fuel mixture. Furthermore, it was discovered that as INHNs and fuel IPs grew, $CO_2$ emissions increased. This has the added benefit of allowing the fuel IP to be increased up to a specific limit. $CO_2$ emissions are reduced at fuel IP 220 bar and INHN-4, owing to better atomization and appropriate combustion. Using the standard INHN-1, increasing the fuel IP to 210 bar increases $CO_2$ emissions from diesel fuel. The $CO_2$ emission increased by 6.16% at fuel IP 210 bar, and by 4.75% at 220 bar, compared to 200 bar fuel IP and INHN-4 lowered the proportion of $CO_2$ emission at standard conditions due to incorrect combustion [4]. $CO_2$ emissions increased as INHN was raised, according to the findings. As a result, fuel atomization increases, resulting in more carbon being burned and more $CO_2$ emissions. When compared to the optimal working fuel IP of 210 bar utilizing INHN-3, $CO_2$ emissions from diesel fuel increased by 4.23% at INHN-4 but decreased by 11.46% at INHN-1.

(e)    Effect of INHNs and Fuel IP on Hydrocarbon Emissions

Figure 17c shows the effects of INHNs and fuel IP on HC emissions at an ideal load situation for WCO biodiesel (B30) blend fuel. We know that low fuel velocity that is insufficient to penetrate the air spray creates HC emissions, resulting in inappropriate air–fuel mixing and a decreased equivalency ratio. The HC emissions are lower at a fuel IP of 210 bar and an INHN-4 injector nozzle due to the enhanced atomization and appropriate combustion. Improved atomization will also result in a shorter igniting time [4]. Unburned HC is formed as a result of incorrect fuel combustion in the combustion chamber. The graphs below show that with 0.25 mm of INHN-4 and B30 fuel, HC production is significantly reduced. When the orifice diameter is reduced, the high temperature is increased, and the fuel is completely burned in the combustion chamber. Because of the harder reaction induced by the decreased temperature in the combustion chamber, which

is caused by poor atomization and evaporation. It can be perceived that the HC emission for diesel fuel decreases considerably with a rise in fuel IP up to 210 bar, from the standard 200 bar. While, with an increase of fuel IP from 210 bar to 220 bar, HC emissions increased slightly. The HC emissions of the operating fuel were increased by 9.49% and 4.57% when it was worked at 200 bar and 220 bar, respectively, compared to 210 bar fuel IP and INHN-4.

(f)     Effects of INHNs and Fuel IP on Oxides of Nitrogen Emissions

Figure 17d shows the influence of INHNs and fuel IPs on $NO_x$ emissions for WCO biodiesel (B30) blend fuel at an ideal load situation. The oxidation of nitrogen at peak combustion temperature produces $NO_x$ emissions [4]. The $NO_x$ emissions were observed to be rising for every INHN operation, with a rise in fuel IP up to a specific limit because of rapid combustion and higher temperatures attained in the cycle, as illustrated in Figure 17d. However, at a fuel IP of 210 bar for INHN-1 and 220 bar for INHN-3 and 4, the $NO_x$ emission starts decreasing. The INHN-4 delivers superior air and fuel mixing and, therefore, greater premixed combustion happens, resulting in a slight rise in $NO_x$ emissions. It can be perceived that with a rise of fuel IP from 200 to 210 bar, for INHN-4 the $NO_x$ emission was increased by 5.43%, while increasing the fuel IP to 220 bar caused the $NO_x$ emission to marginally decrease by 7.76%.

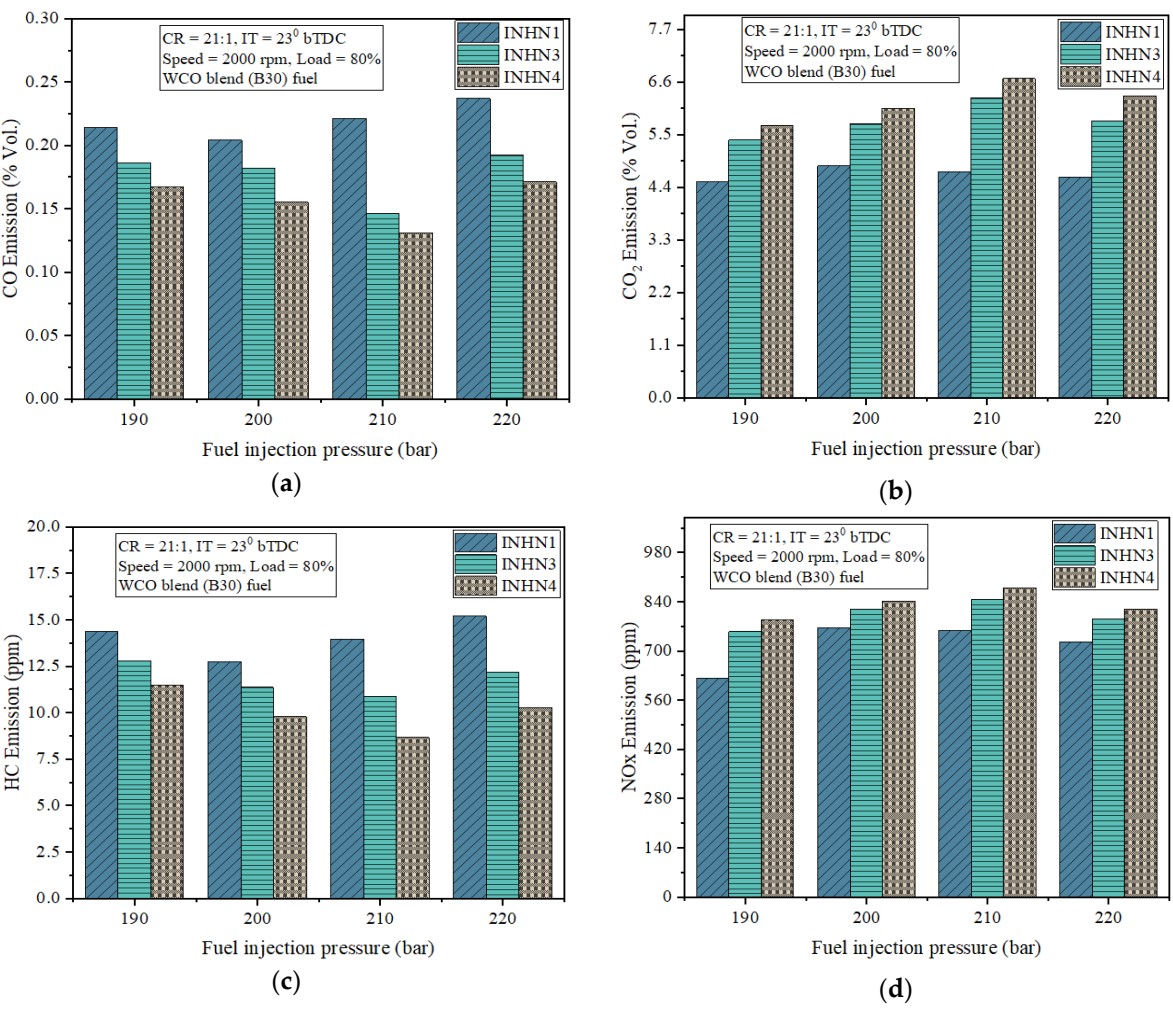

**Figure 17.** Effects of fuel IPs and INHNs on: (**a**) CO; (**b**) $CO_2$; (**c**) HC, and (**d**) $NO_x$ emissions from engine.

## 5. Conclusions

Some of the most critical findings of the numerical simulation are stated as follows:

(a) The cavitation intensity for INHN-4 was higher than for INHN-1 and INHN-3 for the same injection period, according to the nozzle flow simulation findings;

(b) The particle size of the INHN-4 was smaller than that of the INHN-1 due to higher aerodynamic effects in the INHN-4, which might have reduced droplet size, resulting in a larger cloud formation after secondary breakage;

(c) The INHN-4's spray tip penetration was less than the INHN-1's. Because the INHN-1's spray widths and cone angles were smaller, this was the case.

The engine's fuel economy is critical. The following conclusions are formed based on the findings of the experimental investigation:

(a) As the number of injector holes and fuel IPs increased, the BTE also increased, and the BSFC fell until the INHNs and fuel IPs reached a predetermined limit. INHN-4, on the other hand, has a 3.16% greater BTE at 210 bar. As a result, boosting injection pressure and hole count had a significant impact on engine performance;

(b) The CO, $CO_2$, and HC emissions for INHN-3 and INHN-4 at optimum load were determined to be lowest at 210 bar and 200 bar for INHN-1, respectively. The CO emissions for INHN-4 were 5.58%, 18.82%, and 19.68% lower, respectively, while the HC emissions were 8.88%, 17.49%, and 8.59% lower than for INHN-3 at 200 bar, 210 bar, and higher 220 bar fuel IP. When compared to higher pressures of 200 bar, 210 bar, and lower pressures of 220 bar, $NO_x$ emissions were 6.79%, 5.43%, and 7.76%, while $CO_2$ emissions were 6.65%, 10.28%, and 5.80%, respectively;

(c) For WCO biodiesel blend fuel, the BTE and BSFC are improved with increased INHNs and fuel IPs;

(d) With increased INHNs and fuel IPs, CO, and HC exhaust emissions are reduced, whereas $CO_2$ and $NO_x$ emissions are somewhat increased;

(e) Increasing fuel IP from 200 to 210 bar for WCO biodiesel blend fuel with INHN-4 provided the preeminent outcome for the engine performance parameters.

**Author Contributions:** Conceptualization, M.B.A. and M.W.M.; methodology, M.B.A. and M.W.M.; software, M.B.A.; investigation, M.B.A. and M.W.M.; resources, M.B.A. and M.W.M.; data curation, M.B.A.; writing, M.B.A. and M.W.M.; supervision, M.W.M.; project administration M.B.A. and M.W.M.; funding acquisition, M.B.A. and M.W.M. All authors have read and agreed to the published version of the manuscript.

**Funding:** The authors would like to recognize Ethiopian Defence University, College of Engineering.

**Data Availability Statement:** The dataset used in this research are available upon request from the corresponding author.

**Conflicts of Interest:** The authors declare no conflict of interest.

## Nomenclature

| | |
|---|---|
| BP | Brake Power (kW) |
| BSFC | Brake specific fuel consumption (kg/kWh) |
| bTDC | Before top dead center |
| BTE | Brake thermal efficiency (%) |
| CFD | Computational fluid dynamics |
| CI | Compression ignition |
| CR | Compression ratio |
| DI | Direct Injection |
| INHN | Injector nozzle hole number |
| IP | Injection pressure (bar) |
| IT | Injection timing (°CA) |
| NHD | Nozzle hole diameter |
| WCO | Waste Cooking Oil |
| 100D | Neat diesel fuel |

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
