# Peer review of "Effects of Injector Nozzle Number of Holes and Fuel Injection Pressures on the Diesel Engine Characteristics Operated with Waste Cooking Oil Biodiesel Blends"

_2673-3994, doi:10.3390/fuels3020017_

Round 1

Reviewer 1 Report

The current format of the paper is not acceptable to me, the authors should add a new idea to this type of study. The novelty and potential impact of this paper should be clarified. I recommend that you edit the entire article according to journal standards and re-send it. The manuscript is a research technical report, I regret to inform you that it cannot be accepted for publication. They should be able to propose a model to cover a gap based on the literature and they should evaluate the model based on justifications and comparisons concerning those proposed in the literature. Finally, the readers of this paper should be able to get the main points through some results and a summary.

An updated and complete literature review should be conducted to present the state-of-the-art and knowledge gaps of the research with strong relevance to the topic of the paper.

The originality of the paper needs to be further clarified. It is of importance to have sufficient results to justify the novelty of a high-quality journal paper.

The results should be further elaborated to show how they could be used for real applications. Modelling results should be validated by experiments.

Author Response

Response to Reviewer 1 Comments

Point 1: The current format of the paper is not acceptable to me, the authors should add a new idea to this type of study. The novelty and potential impact of this paper should be clarified. I recommend that you edit the entire article according to journal standards and re-send it. The manuscript is a research technical report, I regret to inform you that it cannot be accepted for publication. They should be able to propose a model to cover a gap based on the literature and they should evaluate the model based on justifications and comparisons concerning those proposed in the literature. Finally, the readers of this paper should be able to get the main points through some results and a summary.

Response 1: Based on the recommendations given, we  tried to add a new idea, potential impact of the study according to journal standards at the abstract section. Related literature are reviewed and summarized in comparison to the current work in the introduction section.

Point 2: An updated and complete literature review should be conducted to present the state-of-the-art and knowledge gaps of the research with strong relevance to the topic of the paper.

Response 2: A relevant literature is introduced and the uniqueness of the current study is identified.

Point 3: The originality of the paper needs to be further clarified. It is of importance to have sufficient results to justify the novelty of a high quality journal paper. 

Response 3: Originality of the work is also clarified so that results are satisfactory.

Point 4: The results should be further elaborated to show how they could be used for real applications. Modeling results should be validated by experiments.

Response 4: Results are explained in detail so that it is easy to apply based on the comment given. But due to unavailability of experimental spray character testing machines in Ethiopian Defence University, College of Engineering, modeling results cannot be validated experimentally.

Reviewer 2 Report

I have read the article carefully and I thought you might like a few suggestions:

  • As the given graphs must be visually clear, authors should redraw the graphs of article as they are of poor quality.
  • I would prefer deeper scientific discussion in the discussion of the results. 

Author Response

Response to Reviewer 2 Comments

Point 1: We have read the article carefully and We thought you might like a few suggestions: As the given graphs must be visually clear, authors should redraw the graphs of the article as they are of poor quality.

Response 1: Based on the suggestions given almost all graphs in this paper are done again to be clear and good quality.

Point 2: We would prefer deeper scientific discussion in the discussion of the results.

Response 2: The results in the study are discussed to explain what they are indicating and on base of the comment tried to be scientific.

Reviewer 3 Report

The present study represents a research effort to understand the effects of parameters such as the number of holes in the injector nozzle and injection pressure of the diesel engine performance. The authors put significant efforts to address the research questions in this study. However, the data representation in the article needs a significant amount of work. The following comments can be helpful to improve the quality of this manuscript:

Abstract: Please provide one sentence about what problem was addressed during this study.

Introduction: This section needs a lot of improvement. It contains very limited and confusing information about the topics related to the study. There are a lot of gaps between different pieces of information. Please try to describe the topics that are later used in the manuscript, also, try to summarize the introduction towards what is the research problem, related topics, what is used to solve the problem, and in a sentence what has been achieved so far. Define the WCO, B10, B20, B30 in the introduction, which will help to maintain the flow of the manuscript. Why is the CFD/computational model not introduced in this section?

Figure 1: It is too general, please add more specifications here.
Figure 2 and 3: What are the differences between a, b, and c. Please mention in the figure captions. Please do not leave anything for the readers to interpret.
Figure 4: What are the numbers in legends? This is a poorly annotated figure. b, c and d illustrate the same profiles INHN-1, 2,3, and 4? This figure needs more explanation in the caption for a better understanding of plots. 

Is “Injector nozzle flow phase” the sub-heading of “Mathematical model and governing equations”? If so please make it visible by using numbers or different font sizes.

Table 2: Please define why these values of boundary conditions were used. Also, provide appropriate references if applicable here. 

Table 3: Again, are these values assumptions or learned for the previous studies? Please be more specific. References?

Figure 4: Define each term in the box otherwise this figure does not add any value to the manuscript.   

Eulerian model? Use a few sentences to define such terms to understand the importance of these in the study.

Conclusion: Please concise the text in this section. Many of these observations can move to the results section. 

Author Response

Response to Reviewer 3 Comments

Point 1: The present study represents a research effort to understand the effects of parameters such as the number of holes in the injector nozzle and injection pressure of the diesel engine performance. The authors put significant efforts to address the research questions in this study. However, the data representation in the article needs a significant amount of work. The following comments can be helpful to improve the quality of this manuscript:

Abstract: Please provide one sentence about what problem was addressed during this study.    

Response 1: In the abstract section the basic problems addressed in this study are also included. The basic problem addressed in this study was improved fuel properties,  increasing INHNs and fuel IPs significantly influences blend fuel properties of viscosity and density leads to improved atomization and mixing rates, and combustion and engine efficiency. 

Point 2: Introduction: This section needs a lot of improvement. It contains very limited and confusing information about the topics related to the study. There are a lot of gaps between different pieces of information. Please try to describe the topics that are later used in the manuscript, also, try to summarize the introduction towards what is the research problem, related topics, what is used to solve the problem, and in a sentence what has been achieved so far. Define the WCO, B10, B20 and B30 in the introduction, which will help to maintain the flow of the manuscript. Why is the CFD/computational model not introduced in this section?

Response 2: Based on the comments given the introduction section improved. The topics in the manuscript are defined, related topics based on literature are also introduced here. The waste cooking oil biodiesel blends are defined and specified in section 3.1 Fuel properties. The computational model is also tried to introduce but discussed in detail at section 2 Methodology.

Point 3: Figure 1: It is too general, please add more specifications here.

Response 3: Figure 1 is the general computational CFD analysis flow chart but, almost all the theories and definitions given in section 2 Methodology are about the specific aspects of the modeling and analysis.

Point 4: Figure 2 and 3: What are the differences between a, b, and c. Please mention in the figure captions. Please do not leave anything for the readers to interpret.

Response 4: Based on the comment in Figure 2 and 3 it was tried to discuss the differences and interpreted.

Point 5: Figure 4: What are the numbers in legends? This is a poorly annotated figure. b, c and d illustrate the same profiles INHN-1, 3, and 4? This figure needs more explanation in the caption for a better understanding of plots.

Response 5: In Figure 4 the legends in numbers are the number mesh elements. Even though figure b, c, and are the same profile but they are different mesh elements and results. Their profile indicates they tend to converge and it is acceptable. For better understanding the detailed explanation is also added below Figure 4.

Point 6: Is “Injector nozzle flow phase” the sub-heading of “Mathematical model and governing equations''? If so please make it visible by using numbers or different font sizes.

Response 6: “Injector nozzle flow phase” is the sub-heading of “Mathematical model and governing equations” and corrected by using numbers to be visible.

Point 7: Table 2: Please define why these values of boundary conditions were used. Also, provide appropriate references if applicable here.

Response 7: In section 2.4 based on the comment detail explanations and appropriate reference with a new sample picture of boundary section was included. The values in Table 1 are measured values from the existing injector nozzle single hole (INHN-1) using an injector testing machine at the standard air condition.

Point 8: Table 3: Again, are these values assumptions or learned for the previous studies? Please be more specific. References?

Response 8: These values are measured from the existing injector nozzle single hole (INHN-1).

Point 9: Figure 4: Define each term in the box otherwise this figure does not add any value to the manuscript.

Response 9: The terms in the box of Figure 4 are necessary for coupling of the physics behind the simulation work and tried to be defined in section 2.5(b).

Point 10: Eulerian model? Use a few sentences to define such terms to understand the importance of these in the study.

Response 10: Eulerian model is used to model continuous flow of particles based on the Navier-Stokes equation without capturing the discrete particles. This has an advantage to capture the flow inside the injector nozzles and again to couple the flow from the nozzles to discrete particles inside the spray domain using discrete particle model.

Point 11: Conclusion: Please concise the text in this section. Many of these observations can move to the results section.

Response 11: Based on the comment given the conclusion section was concised and tried to summarize the findings of the study.

Round 2

Reviewer 1 Report

The manuscript quality should be improved reporting more detailed analysis and improving the scientific soundness. However, the authors have improved the manuscript compared to the submitted draft.

Author Response

Rebuttal Manuscript (Fuels-1586117)

We appreciate all the reviewer’s comments. We have tried to give responses to the reviewer’s comments thoroughly one by one and the corrections and changes have been made in the revised manuscript.

Response to Reviewer 1 Comments

Point 1: Language usage throughout the manuscript needs to be carefully checked.

Response 1: Language usage throughout the manuscript has been checked and incorporated. into the revised manuscript. Specifically, extensively checked in: Abstract, Introduction, Methodology and Result and discussion part.

Point 2: The manuscript quality should be improved reporting more detailed analysis and improving the scientific soundness. However, the authors have improved the manuscript compared to the submitted draft.

Response 2: Thank you for your further comments you have given to improve the manuscript quality. Accordingly the detailed analysis of the simulation work is further illustrated for clear understanding of readers on the base of scientific analysis reference of related works. Some graphs are added with their explanations in improving the analysis of the manuscript in detail. Specifically, the detail of internal nozzle flow simulation at section 4.1(a) and graphical representation and detailed explanation of the spray simulation (spray width, particle penetration, particle velocity and particle mean diameter are expressed more.

Reviewer 3 Report

The authors addressed previous comments very carefully. Therefore, I do not have any further major comments.

Figure 1 - The line thickness in (a) and (b) panels is inconsistent.

Author Response

Rebuttal Manuscript (Fuels-1586117)

We appreciate all the reviewer’s comments. We have tried to give responses to the reviewer’s comments thoroughly one by one and the corrections and changes have been made in the revised manuscript.

Response to Reviewer 3 Comments

Point 1: The authors addressed previous comments very carefully. Therefore I don’t have any further comments.

Figure 1- The line thickness in (a) and (b) panels is inconsistent.

Response 1: Thank you for the continuous review and comments you have given. Accordingly, the authors tried to understand the figure number indication error. And the line thickness inconsistency in panels of Figure 4(b), (c) and (d) is corrected and made to be unique in line thickness as of Figure 4(a) instead.
